# Mechanism of Intracellular Elemental Sulfur Oxidation in *Beggiatoa leptomitoformis*, Where Persulfide Dioxygenase Plays a Key Role

**DOI:** 10.3390/ijms252010962

**Published:** 2024-10-11

**Authors:** Tatyana S. Rudenko, Liubov I. Trubitsina, Vasily V. Terentyev, Ivan V. Trubitsin, Valentin I. Borshchevskiy, Svetlana V. Tishchenko, Azat G. Gabdulkhakov, Alexey A. Leontievsky, Margarita Yu. Grabovich

**Affiliations:** 1Department of Biochemistry and Cell Physiology, Voronezh State University, 394018 Voronezh, Russia; 2Federal Research Center “Pushchino Scientific Center for Biological Research of the Russian Academy of Sciences”, G.K. Skryabin Institute of Biochemistry and Physiology of Microorganisms of the Russian Academy of Sciences, 142290 Pushchino, Russia; 3Federal Research Center “Pushchino Scientific Center for Biological Research of the Russian Academy of Sciences”, Institute of Basic Biological Problems, 142290 Pushchino, Russia; 4Research Center for Molecular Mechanisms of Aging and Age-Related Diseases, Moscow Institute of Physics and Technology, 141701 Dolgoprudny, Russia; 5Institute of Protein Research, Russian Academy of Sciences, 142290 Pushchino, Russia

**Keywords:** filamentous colorless sulfur bacteria, *Beggiatoa leptomitoformis*, sulfur metabolism, elemental sulfur oxidation, persulfide dioxygenase

## Abstract

Representatives of the colorless sulfur bacteria of the genus *Beggiatoa* use reduced sulfur compounds in the processes of lithotrophic growth, which is accompanied by the storage of intracellular sulfur. However, it is still unknown how the transformation of intracellular sulfur occurs in *Beggiatoa* representatives. Annotation of the genome of *Beggiatoa leptomitoformis* D-402 did not identify any genes for the oxidation or reduction of elemental sulfur. By searching BLASTP, two putative persulfide dioxygenase (PDO) homologs were found in the genome of *B. leptomitoformis*. In some heterotrophic prokaryotes, PDO is involved in the oxidation of sulfane sulfur. According to HPLC-MS/MS, the revealed protein was reliably detected in a culture sample grown only in the presence of endogenous sulfur and CO_2_. The recombinant protein from *B. leptomitoformis* was active in the presence of glutathione persulfide. The crystal structure of recombinant PDO exhibited consistency with known structures of type I PDO. Thus, it was shown that *B. leptomitoformis* uses PDO to oxidize endogenous sulfur. Additionally, on the basis of HPLC-MS/MS, RT-qPCR, and the study of PDO reaction products, we predicted the interrelation of PDO and Sox-system function in the oxidation of endogenous sulfur in *B. leptomitoformis* and the connection of this process with energy metabolism.

## 1. Introduction

Representatives of the genus *Beggiatoa* belong to the group of filamentous colorless sulfur bacteria. These microorganisms are found in biotopes where reduced sulfur compounds such as hydrogen sulfide or thiosulfate are usually present. Presently, the genus includes two species of heterotrophic freshwater representatives, *Beggiatoa alba* and *Beggiatoa leptomitoformis*, which are able to grow lithotrophically by oxidizing reduced inorganic sulfur compounds [1,2]. Bacteria use these compounds as electron donors for energy metabolism. As a result of their oxidation, globules of elemental sulfur accumulate in the invaginates of the cytoplasmic membrane of these bacteria [1,2,3,4,5]. For representatives of the genus *Beggiatoa*, the mechanism of further conversion of accumulated sulfur into sulfate is not shown [6,7]. Although, previously Schmidt et al. [6] suggested that under anaerobic conditions *B. alba* can use intracellular elemental sulfur as a terminal electron acceptor, as sulfide accumulation was observed during short-term culture under anaerobic conditions. However, there is still no conclusive evidence for this process.

An interesting fact is that when *Beggiatoa* is cultured on a medium without an exogenous sulfur source under aerobic conditions, the cells begin to lose accumulated sulfur globules, but how this process is realized has not yet been studied.

The best known enzyme systems that are involved in the dissimilatory conversion of elemental sulfur to sulfite in bacteria are the reverse dissimilatory sulfite reductase rDsr pathway and the sulfur-oxidizing heterodisulfide reductase-like sHdr pathway. Both pathways are widely distributed in the domain of Bacteria among chemotrophic and phototrophic sulfur-oxidizing prokaryotes, and the sHdr pathway is also found in archaea [8,9].

Oxidation of sulfane sulfur can also be carried out by persulfide dioxygenase (PDO) from the superfamily of metallo-β-lactamase enzymes [10]. However, PDO does not oxidize elemental sulfur directly because its direct substrate is glutathione persulfide (GSSH) [10,11]. There is a mechanism for the oxidation of hydrogen sulfide to sulfate involving sulfide:quinoxidoreductase (SQR) and PDO, which is used by many heterotrophic and chemoautotrophic bacteria to detoxify hydrogen sulfide [12,13,14,15,16]. In this process, SQR oxidizes sulfide to sulfane sulfur (S^0^/polysulfide), which then reacts with reduced glutathione (GSH) to form GSSH [17]. Rhodanese can accelerate the reaction of sulfane sulfur with GSH [17,18]. Then, PDO oxidizes GSSH to sulfite according to the following reaction [10,12,19]:GSSH + O_2_ + H_2_O → GSH + SO_3_^2−^ + 2H^+^

The sulfite formed is able to spontaneously react with sulfane sulfur to form thiosulfate [15], which is then oxidized to sulfate with the participation of the Sox-system.

Currently, persulfide dioxygenases are categorized into three types [12,14]. Type I includes known PDOs of human (hETHE1, pdb id 4CHL), *Arabidopsis thaliana* (pdb id 2GCU), and many bacterial PDOs, including PDO from *Paraburkholderia phytofirmans* (*Bp*PRF, pdb id 5VE5), which fuses a C-terminal region with a rhodanese domain, and *Myxococcus xanthus* (*Mx*PDO, pdb id 4YSB) [13,20,21,22]. Type II is widely distributed in *Pseudomonadota* (formerly *Proteobacteria*), e.g., PDO of *Pseudomonas putida* (PpPDO2, pdb id 4YSL) [13]. Type III PDOs are characterized by the presence of a rhodanese domain similar to *Bp*PRF, but these enzymes differ in substrate specificity [23]. In enzymes of the metallo-β-lactamase superfamily, the dominant form of metal in the active site is Fe^2+^. However, some type II PDOs can also use Mn^2+^ for catalysis, and glyoxalase II proteins can contain two metal ions in the active site at once [12].

The present study reports a putative mechanism for the oxidation of intracellular elemental sulfur in *B. leptomitoformis* involving persulfide dioxygenase similar to that described for heterotrophic microorganisms, and shows for the first time its link to energy metabolism. Here, we report the first structure of a recombinant PDO obtained for a representative from a group of colorless filamentous sulfur bacteria, *Beggiatoa leptomitoformis*, and provide a comparative structural and biochemical characterization with known PDOs.

## 2. Results

### 2.1. Genome Annotation and Phylogenetic Analysis

Previous bioinformatic sequence analysis of the D-401 and D-402 genomes revealed that various enzymes and enzyme complexes located in different cellular compartments are involved in the conversion of sulfur compounds in *B. leptomitoformis* [1,2] (Figure 1).

The genomes contain genes of the system of oxidation hydrogen sulfide SqrQ, sulfide:quinone oxidoreductase type I (*sqrA*) and type VI (*sqrF*) and FCSD, flavocytochrome sulfide dehydrogenase (*fccB*). Also, the genomes of the two strains encode genes for the direct oxidation of sulfite to sulfate, the SoeABC complex, membrane-bound cytoplasmic sulfite:quinone oxidoreductase (*soeABC*), and genes of the branched Sox-system without SoxCD proteins (*soxAXBYZ*) for the oxidation of thiosulfate to sulfur and sulfate [1,2].

Additional analysis of the genomes of *B. leptomitoformis* strains showed the presence of genes for the sulfite exporter TauE/SafE family protein, rhodanese (Rhd), DsrE, and TusA.

However, a search for the conversion of intracellular stored sulfur did not reveal any genes for intracellular sulfur-oxidation enzymes such as reverse dissimilatory sulfite reductase (rDsrAB) and the sulfur-oxidizing heterodisulfide reductase-like pathway (sHdr), nor any genes for the elemental sulfur reduction enzyme sulfur reductase SOR. A BLASTP search using the known protein sequence of persulfide dioxygenase from the NCBI database revealed protein sequences in the strain D-402 with inventory numbers ALG68607.1 and WP_201800136.1 and 30% sequence identity.

Phylogenetic analysis based on the functionally characterized sequences of proteins from this family was conducted to ascertain the functions of the detected proteins [12,14]. Phylogenetic analysis showed clear clustering of the derived sequence ALG68607.1 with persulfide dioxygenase type I sequences. The detected sequence with inventory number WP_201800136.1 clusters with glyoxalase II proteins (Figure 2).

Using the SignalP v.5.0 server, it was revealed that the amino acid sequence of ALG68607.1 does not contain signal peptides or transmembrane regions. This suggests that this protein may be a cytoplasmic protein similar to other bacterial PDOs. The cytoplasmic localization of the detected PDO homolog from *B. leptomitoformis* was predicted using the PSORTB v3.0 tool.

In some prokaryotes, persulfide dioxygenase is involved in the oxidation of sulfane sulfur [13,14,15,22,23]. Therefore, it was suggested that persulfide dioxygenase may potentially participate in the process of transformation of intracellular stored sulfur in representatives of the genus *Beggiatoa*.

### 2.2. Proteomic Analysis of B. leptomitoformis Culture Grown in the Presence of Endogenous Sulfur and CO_2_ Only

To prove the PDO function in *B. leptomitoformis,* proteomic analysis of the culture was performed by HPLC-MS/MS. Analysis was performed with the strain D-402 grown on starvation mineral medium with endogenous sulfur as a potential electron donor for energy metabolism and CO_2_ as a carbon source. The experimental sample was taken on the seventh day of bacterial incubation on starvation mineral medium. The incubation time was calculated based on the maximum result of the *pdo* gene expression level obtained by RT-qPCR analysis (in the section Quantitative Analysis of *pdo* and *soxAXBYZ* Gene Expression). The cultures of the strain D-402 grown under chemolithoheterotrophic conditions in the presence of thiosulfate and lactate and organoheterotrophically in the presence of lactate were used as control samples.

The analysis provided data on the reliable presence of 1989 proteins in the experimental and control samples. Proteins were identified by at least two peptides, the sequence of each of which was deciphered from the tandem mass spectrometric spectrum. The presence of major proteins involved in hydrogen sulfide metabolism to sulfate in *B. leptomitoformis* D-402 was evaluated separately in the samples. Analysis of protein occurrence and signal intensity allowed us to identify the persulfide dioxygenase PDO (TrEMBL identifier UPI000706C17D) only in the culture sample grown on starvation mineral medium in the presence of endogenous sulfur, suggesting that this protein in *B. leptomitoformis* is differentially expressed. For SoxB proteins (identifier in TrEMBL UPI0007061722) and the transport proteins DsrE (identifier in TrEMBL UPI00070692D2) and TusA (identifier in TrEMBL UPI0007063E4B), a statistically significant change in expression level was detected in the culture sample also grown in the presence of endogenous sulfur.

For SQR of types I and VI, FCSD, the Sox-system, and TauE/SafE transporter proteins, there was no statistical difference in the expression of these proteins between control and experimental samples. SoeABC proteins could not be identified in any of the samples which excludes the possibility that this complex oxidizes sulfite formed as a result of the persulfide dioxygenase reaction. It is worth noting that SoxAXYZ proteins did not show a statistically significant change in expression level. Nevertheless, the detection of these proteins in control and experimental samples may indicate that they are constitutive in *B. leptomitoformis* D-402. This is thought to be due to the features of the ecological niches in which these microorganisms prevail. Typically, such biotopes are characterized by the content of hydrogen sulfide and thiosulfate, which *Beggiatoa* spp. use in the processes of energy metabolism [24,25,26]. Thiosulfate is often metabolized in addition to sulfide as an energy substrate.

The lack of differences in the expression profiles of some proteins can be explained by their function in the cell. The SQR and FCSD proteins catalyze the oxidation reaction of hydrogen sulfide, which is accompanied by the accumulation of elemental sulfur inside the cell. The growth conditions for analysis did not include any sulfur sources other than endogenous sulfur, and thus the lack of substrate for the SQR and FCSD enzymes could be the reason for the identical expression profiles.

### 2.3. Cloning of the PDO Gene, Its Expression and Purification of Protein, and the Physicochemical Properties of Recombinant PDO

To confirm the involvement of PDO in a sulfur-oxidation process in *B. leptomitoformis,* the gene encoding persulfide dioxygenase was cloned from the D-402 strain into the pET-22b(+) plasmid. Three conserved amino acids of the iron-binding site (His57, His113, and Asp134) and three conserved amino acids of the substrate-binding pocket (Asp61, Tyr177, and Arg194) were identified in the sequence. These residues are conserved and present in other type I PDOs in humans (hETHE1, pdb id 4CHL), *Arabidopsis thaliana* (pdb id 2GCU), *Myxococcus xanthus* (*Mx*PDO, pdb id 4YSB), and *Paraburkholderia phytofirmans* (*Bp*PRF, pdb id 5VE5) [13,20,21,22]. The *pET22b::pdoI* construct was used for the transformation of competent cells of *E. coli* BL21-CodonPlus. *E. coli* cells carrying the recombinant vector produced PDO in a high yield, about 12 mg per liter of culture. Persulfide dioxygenase was expressed as a mature enzyme with a 6 × His-tag at the N-terminus instead. Purification was carried out in one step using affinity chromatography. Based on the amino acid sequence of PDO, the calculated molecular mass of the protein was 28.6 kDa. The enzyme boiled with β-mercaptoethanol and SDS migrated at 29 kDa in SDS-PAGE, which corresponded to the theoretically calculated value of molecular mass (Figure 3).

It is known that sulfide is oxidized to form sulfane sulfur, which is then transferred to glutathione to form glutathione persulfide (GSSH), which is the substrate for the persulfide dioxygenase reaction [10,27]. To verify the ability of the obtained recombinant PDO to oxidize the canonical substrate, the enzyme activity was measured by the rate of oxygen uptake in the presence of GSSH. In 50 mM potassium phosphate (KPi) buffer (pH 7.0) at 35 °C, the rate of oxygen uptake due to enzyme activity was ~174 μmol O_2_ mg^−1^ h^−1^ which corresponded to ~2900 U mg^−1^ (Figure 4a).

The study of the pH optima of PDO activity revealed an optimum in the range of 7.0–7.5. At the same time, pH < 7.0 critically inhibited the enzyme activity in contrast to pH > 7.8. Thus, while at pH 6.5 the activity was reduced by ~75% of the maximum, at pH 8.0 it was reduced by only 10–13% (Figure 4a). The temperature optimum of the obtained protein was characterized by a bell-shaped curve with a clear peak at 40 °C where the oxygen uptake rate was ~190 μmol O_2_ mg^−1^ h^−1^ or ~3167 U mg^−1^ (Figure 4b).

The thermal stability of PDO was determined by incubating the enzyme in 50 mM KPi buffer at different temperatures over the range of 30 °C to 50 °C. After one hour of incubation at 30 °C, the protein retained 84% of its initial activity. At incubation temperatures of 40 °C, 45 °C, and 50 °C, PDO activity decreased mainly in the first 15 min and after one hour it was 64%, 16%, and 2%, respectively (Figure 5).

The incubation of PDO at different pH values showed the following relationship (Figure 6). At the beginning of the test, PDO retained maximum initial activity after incubation of the protein in buffers with pH throughout the tested range (6.0–11.0) and subsequent measurement of residual activity in 50 mM KPi buffer (pH 7.0) at 35 °C. However, its activity was strongly lost at pH 11.0 (by ~55%). On the second day of incubation, the enzyme activity strongly decreased at pH 10.0. As a result, the PDO activity values at pH 10.0 and pH 11.0 were no more than 10–13% of that observed at pH 9.0, while at other pH values the loss of activity did not exceed 10–12% compared to day 0. On day 4, enzyme activity began to decline markedly at pH 6.0 (~55% compared to that observed at pH 7.0). On day 7, the maximum enzyme activity of PDO was observed at pH 7.0 and 8.0 (30–33% decline from day 0). Medium activity was observed at pH 6.0 and 9.0 (~57% and ~51% decline in activity from day 0, respectively). At pH 10.0–11.0, the enzyme activity of PDO was lost completely.

A study of the inhibitory effect of metal ions and EDTA on recombinant PDO activity showed that Cu^2+^, Zn^2+^, Co^2+^, Ni^2+^, and Fe^3+^ at a concentration of 1 mM significantly inhibited PDO activity, while Mg^2+^ ions had no effect on enzyme activity and Mn^2+^ ions reduced activity to only ~83% (Table 1). The addition of EDTA did not lead to a significant change in activity. When 1 mM EDTA was used, 92.5% of recombinant PDO activity was retained.

### 2.4. Structure of PDO from B. leptomitoformis

The *B. leptomitoformis* PDO structure reveals dimeric organization and contains an αββα metal-binding fold with two central mixed β-sheets, each containing six strands, surrounded on both sides by helices (Figure 7a). In the *B. leptomitoformis* PDO structure, 14 residues from the C-terminus are not visible, which indicates the mobility of this region. One molecule of the *B. leptomitoformis* PDO dimer is located in the asymmetric unit.

The PDO of *B. leptomitoformis* shows a high overall similarity to the known type I PDO structures of humans (hETHE1, pdb id 4CHL; RMSD = 1.1 Å), from *Arabidopsis thaliana* (pdb id 2GCU; RMSD = 1.07 Å), and *Myxococcus xanthus* (*Mx*PDO, pdb id 4YSB; RMSD = 0.97 Å), and the PDO–rhodanese fusion from *Paraburkholderia phytofirmans* (*Bp*PRF, pdb id 5VE5; RMSD = 1.05 Å) [13,20,21,22]. Although the PDO structures are similar, there are some differences.

The dimerization interface for PDO *B. leptomitoformis* involves surfaces containing the C-terminal regions of the monomers, while in the dimers of hETHE1 (Figure 7b), as well as *Arabidopsis thaliana* PDO (pdb id 2GCU) and *Mx*PDO, the dimerization surfaces included N-terminal regions [13,21]. In addition to the arrangement of monomers, the areas of intermolecular interfaces also differ. Thus, in PDO from *B. leptomitoformis*, the interface area between monomers is not extensive (997 Å^2^), as also seen in the similar area in the structure of PDO from *A. thaliana* [21]. The total buried surface area at the interface in hETHE1 is 2-fold larger (1950 Å^2^) [20]. There are 10 hydrogen bonds involved in the formation of contacts between *B. leptomitoformis* PDO monomer molecules, and there is a hydrophobic interaction.

In the active site of PDO from *B. leptomitoformis* (Figure 8), the residues of the first coordination sphere of the iron ion (His57, His113, Asp134, and three water molecules) are identical to those of other type I PDO structures [13,20,21,28]. The substrate in the active site of the *B. leptomitoformis* PDO can be coordinated by an iron ion, Arg194, Tyr177, and Asp61 [20,21,29,30].

### 2.5. Quantitative Analysis of pdo and soxAXBYZ Gene Expression

Long-term observations (25 days) of *B. leptomitoformis* D-402 during chemolithoautotrophic growth in the presence of intracellular sulfur revealed with regularity that in the first 7 days, endogenous sulfur gradually disappears from the cell and then accumulates again.

To test the involvement of the detected persulfide dioxygenase protein in elemental sulfur transformation processes in *B. leptomitoformis* D-402, the *pdo* gene expression was analyzed by relative RT-qPCR during chemolithoautotrophic growth of the strain where intracellular sulfur acted as an electron donor for energy metabolism. Strain D-402 grown organoheterotrophically on lactate and chemolithoautotrophically in the presence of sulfide as an electron donor for energy metabolism was used as a control.

We assessed the expression level of *pdo* at time points that were distributed to cover the period of reduction in the amount of sulfur globules in the cell. The analysis showed that maximum expression of the *pdo* gene in *B. leptomitoformis* was reached on the seventh day of incubation in the presence of intracellular elemental sulfur alone (Figure 9A). The expression of *pdo* gene increased an average of 35-fold over growth at other time points, and an average of 25-fold over control growth conditions (Figure 9A).

Given the fact of re-accumulation of intracellular sulfur globules, the expression level of Sox-system genes was measured at time points covering the phase of intracellular sulfur re-accumulation.

The maximum gene expression was reached on the 18th day of incubation under experimental conditions. Expression of the *soxAX* and *soxB* genes was on average 15-fold higher, and *soxY* gene expression was 8.6-fold higher during growth in the presence of endogenous sulfur alone compared to chemolithoautotrophic growth in the presence of sulfide. The expression of Sox-system genes in D-402 during organoheterotrophic growth was minimal and equaled the instrument error (Figure 9B).

This result indirectly confirms that a mechanism can be realized similar to other heterotrophic sulfur-oxidizing bacteria [15]. When sulfite spontaneously reacts with sulfane sulfur, thiosulfate is formed, which is then probably transported into the periplasm with the participation of unknown transporter proteins. In the periplasm, there occurs its oxidation by the branched Sox-system with the formation of elemental sulfur and sulfate. This provides clarity and may explain the re-accumulation of intracellular elemental sulfur in the strain D-402 (in the section Intermediates during cultivation of *B. leptomitoformis* D-402 on starvation medium in the presence of endogenous sulfur). It is important to highlight that the latter reaction for *B. leptomitoformis* is an energy-generating reaction where thiosulfate is an electron donor for energy metabolism.

### 2.6. Intermediates during Cultivation of B. leptomitoformis D-402 on Starvation Medium in the Presence of Endogenous Sulfur

In support of the above hypothesis, we investigated the intermediates during the growth of *B. leptomitoformis* D-402 cultured chemolithoautotrophically in the presence of only intracellular sulfur and CO_2_. We evaluated intermediates during the chemolithoautotrophic growth of *B. leptomitoformis* D-402 in the presence of intracellular sulfur at time points that were distributed to cover the period of reduction in the amount of sulfur globules in the cell and the phase of re-accumulation of intracellular sulfur.

During the chemolithoautotrophic growth of D-402 in the presence of sulfide as an electron donor, the cells accumulated elemental sulfur intracellularly over 3 days, and the increase in protein was 24 mg/L. Intracellular sulfur can account for up to 70% of cell dry weight, as has been shown for *Beggiatoa* spp. and purple sulfur bacteria [31,32]. Then, when the inoculum was transferred to starvation mineral medium, we observed a gradual decrease in elemental sulfur in the cells after 7 days of growth almost to its complete disappearance and an increase in protein up to 28 mg/L. The expression of the *pdo* gene was also maximal on the seventh day of incubation of the strain in the presence of intracellular sulfur as the only energy source.

Notably, after 14 days of incubation under these conditions, a re-appearance of intracellular sulfur globules was observed in the strain D-402. By 24 days of incubation, there was a gradual decrease in the number of viable cells, and the amount of newly formed sulfur globules did not change.

Although the oxidation of sulfane sulfur with PDO participation results in the formation of sulfite, the only other products detected besides intracellular sulfur were thiosulfate and sulfate (Figure 10). Considering the fact that thiosulfate and sulfate are formed intracellularly, we observed the accumulation of intermediates in low concentrations.

## 3. Discussion

Representatives of the genus *Beggiatoa* use oxidation reactions of reduced sulfur compounds to obtain energy. As a result of these reactions, elemental sulfur, which accumulates in the cell, and sulfate are formed as products [1,2,6,33]. The way in which the further transformation of intracellular sulfur is carried out in members of the genus was unknown and remained a mystery for decades.

Observation of the strain *B. leptomitoformis* D-402 during growth in the presence of reduced sulfur compounds showed that the cells obviously lose intracellular sulfur and, more interestingly, re-accumulate it. While other sulfur-oxidizing prokaryotes use sHdr and rDsr to oxidize elemental sulfur, the genome of *B. leptomitoformis* lacks these genes, as well as genes encoding the sulfur reductase SOR of the elemental sulfur reduction. Thus, it is not clear how the disappearance of intracellular sulfur inclusions in representatives of the genus *Beggiatoa* occurs.

In this connection, the question of elemental sulfur oxidation in representatives of the genus *Beggiatoa* remained open for a long time. A BLASTP search identified two protein sequences (ALG68607.1 and WP_201800136.1) in *B. leptomitoformis* D-402 presumably related to persulfide dioxygenase from the metallo-β-lactamase superfamily. Phylogenetic analysis showed clustering of one of the proteins found with type I PDO (Figure 2).

To date, there is no information on the role of this enzyme for members of the genus *Beggiatoa*. However, a sulfide oxidation pathway involving persulfide dioxygenase is described for the heterotrophic bacterium *Cupriavidus pinatubonensis* JMP134 [15,34]. The initial oxidation of sulfide occurs in the cytoplasm under the action of SQR. The resulting sulfane sulfur (S^0^ and polysulfide) is metabolized to sulfite by the action of persulfide dioxygenase. In turn, sulfite is able to chemically react with sulfane sulfur to form thiosulfate, which is then transported to the periplasm and oxidized to sulfate by the Sox-system. This mechanism in heterotrophic prokaryotes is primarily aimed at the detoxification of hydrogen sulfide. However, it should be noted that, to date, persulfide dioxygenase is the only enzyme found that could potentially be involved in the oxidation of stored elemental sulfur in representatives of *Beggiatoa,* such as *B. leptomitoformis* and *B. alba*.

The SQR and FCSD genes of the two systems for the oxidation of sulfide to elemental sulfur are encoded in the D-402 genome. FCSD proteins are soluble periplasmic enzymes, while SQRs can be localized in both periplasm and cytoplasm but are always membrane bound [35]. The use of hydrogen sulfide as an energy substrate during lithotrophic growth requires the presence of multiple, mutually complementary enzymes with well-defined optima of hydrogen sulfide concentration, but which enzyme is preferentially used for sulfide oxidation by *B. leptomitoformis* is not yet clear.

Also, genes for a number of sulfur compound transporters, Rhd, DsrE, and TusA, are present in the genome. In most chemolithoautotrophic sulfur-oxidizing bacteria the *rhd*, *dsrE*, and *tusA* genes are located directly with the *rdsr* and *shdr* genes of intracellular elemental sulfur oxidation. For the purple sulfur-oxidizing bacterium *Allochromatium vinosum,* it was shown that Rhd, TusA, and DsrE are involved in the cytoplasmic transport of persulfide intermediates to the sHdr and rDsr complex in the dissimilatory sulfur-oxidation reaction [36,37,38]. In some cases, the *dsrE*–*rhd*–*tusA* cluster is located immediately upstream of the *dsr* gene cluster, or *tusA* and *dsrE* are linked to the *hdr* genes, or *tusA* is located next to the *dsr* and *soeABC* genes [39,40].

In *B. leptomitoformis,* genes encoding enzymes of dissimilatory sulfur metabolism do not occur in the same cluster but in different parts of the genome. However, in the absence of *rdsr* and *shdr* genes in *B. leptomitoformis*, it is not completely clear what function Rhd, TusA, and DsrE could play in these bacteria. It is possible that in *B. leptomitoformis* these proteins are similarly involved in the transfer of sulfane sulfur through a cascade mechanism with the formation of GSSH, which is oxidized by PDO to sulfite [16,22,38]. In this case, GSH and sulfite can act as sulfane sulfur acceptors [10,14,16].

For *Beggiatoa* spp., it has been shown that sulfur arranged into sulfur globules is in the form of a combination of cyclooctasulfur and inorganic polysulfides [7]. Polysulfides are a versatile pool of bioavailable sulfur. However, cyclooctasulfur requires activation to convert it to the available glutathione persulfide form, which is formed by the non-enzymatic reaction of GSH and elemental sulfur, as shown for representatives of *Acidithiobacillus* and *Acidiphilium* [10]. This produces glutathione persulfide, which is a substrate for persulfide dioxygenase [10].

Proteomic data obtained for the strain D-402 under growth conditions in the presence of endogenous sulfur alone showed a statistically significant change in expression profiles for PDO, SoxB, DsrE, and TusA proteins. SoxAXYZ proteins did not show a statistically significant change in expression levels. However, quantitative RT-PCR data showed an average 6- and 15-fold increase in the expression of the *pdo* persulfide dioxygenase gene and Sox-system genes, respectively, under chemolithoautotrophic conditions in the presence of elemental sulfur alone compared with chemolithoautotrophic growth in the presence of sodium sulfide (Figure 9).

Despite the fact that sulfite is the final intermediate of elemental sulfur oxidation with the participation of persulfide dioxygenase, it could not be registered among the products. According to data for other heterotrophic bacteria, the absence of sulfite among the detected products seems to indicate that the released sulfite does not accumulate in the medium but reacts immediately non-enzymatically with sulfane sulfur to form thiosulfate [13,14,17,41]. 

The gene of the membrane-bound cytoplasmic sulfite:quinone oxidoreductase SoeABC complex is encoded in the genome of *B. leptomitoformis*. Thus it can be assumed that the complex may be involved in sulfite oxidation following the persulfide dioxygenase reaction. However, according to previously obtained data for *B. leptomitoformis*, SoeABC exhibits its activity only under microaerobic conditions, which excludes the possibility of this enzyme functioning in the investigated process, which is carried out under aerobic conditions [2]. These conclusions are confirmed by proteomic analysis, for which culture samples were grown under aerobic conditions, where we were unable to detect any of the proteins of the SoeABC complex.

During the growth of the *B. leptomitoformis* strain, the re-appearance of intracellular sulfur inclusions was observed, and the only products detected other than intracellular sulfur were thiosulfate and sulfate. A similar pattern could be observed if the Sox-system is involved. The genes of the branched Sox-system are encoded in the genome of *B. leptomitoformis*, in which thiosulfate is oxidized to form sulfate and elemental sulfur, which accumulates in cells. Thus, thiosulfate is most likely formed by secondary chemical interaction of sulfite and sulfane sulfur in *B. leptomitoformis*. It is subsequently oxidized in the periplasm with the participation of the branched Sox-system, without SoxCD proteins. This produces elemental sulfur, which re-accumulates in the cell, and sulfate, which we observed as one of the products during the growth of *B. leptomitoformis* in the presence of endogenous sulfur alone. This concept is also confirmed by the results of RT-qPCR and proteomic analysis. However, we did not observe a tendency for thiosulfate to oxidize relative to accumulated sulfate. Probably, this may be due to the fact that the processes of thiosulfate formation and its oxidation through the Sox-system are spatially separated (thiosulfate formation occurs in the cytoplasm and its oxidation in the periplasm).

The formation and further dissimilatory conversion of thiosulfate is crucial for *B. leptomitoformis* because persulfide dioxygenase catalyzes the reaction without generating energy. At the first view, the persulfide dioxygenase reaction seems to be an unnecessary loss to the cell. We should take into account the fact that the strain D-402 grew well when using intracellular elemental sulfur as the only energy source and CO_2_ as an inorganic carbon source. That is, thiosulfate here may act as an indirect electron donor for energy metabolism, being oxidized by the dissimilatory pathway through the Sox-system.

Structural and Biochemical Characterization of Recombinant PDO from *B. leptomitoformis* D-402.

The physicochemical properties of *B. leptomitoformis* PDO were similar to those of other previously studied enzymes of this family. PDOs were characterized by maximum activity at the physiological pH of the cytoplasm (from 7.4 to 7.8) [12]. In this study, the optimal pH for GSSH oxidation was pH 7.5, which agrees well with the previously obtained data. The enzyme demonstrated maximal activity at 40 °C, which also agrees well with previously obtained data for other enzymes (usually PDO enzymes show maximal activity at 35–45 °C) [11,12,28,42]. The enzyme was not characterized by high thermal stability. A decrease in PDO activity was observed after one hour of incubation at 30 °C. This may be due to the fact that the strain producing the enzyme is a mesophilic strain that lives at temperatures ranging from 8 °C to 32 °C, and, therefore, it is not necessary to produce thermally stable enzymes for normal metabolic processes.

The study of the effect of inhibitors on PDO activity showed that Cu^2+^, Zn^2+^, and Ni^2+^ at a concentration of 1 mM significantly reduce the enzyme activity, Mn^2+^ inhibited the PDO to a much lesser extent, while Mg^2+^ and EDTA reduced the activity marginally. Several studies have also reported the inhibitory effect of Cu^2+^, Zn^2+^, and Ni^2+^ on PDO activity [11,42]. The inhibitory effect of Cu^2+^, Zn^2+^, and Ni^2+^ may be related to the substitution of an iron ion in the active site of the PDO.

The crystal structure of PDO *B. leptomitoformis* allowed us to perform a comparative structural analysis with the active sites of other type I PDOs. We compared the putative substrate-binding PDO pocket of *B. leptomitoformis* with corresponding sites in hETHE1, *Mx*PDOI, and *Bp*PRF in complex with glutathione. 

The amino acid residues of the putative substrate-binding pocket of the *B. leptomitoformis* PDO, as in other type I PDOs, form two walls, one structurally rigid, conserved wall formed by, among others, Arg194 and Tyr177, and a second wall opposite, a structurally mobile wall formed by two loops (Figure 11 and Figure 12). One of these loops (containing Thr213, Asn215, and Arg217) is more variable (Figure 11a), both in shape (Arg217 in the PDO from *B. leptomitoformis* and *Bp*PRF is oriented in the other direction unlike Lys216 in hETHE1) and charge (Asn215 and Thr213 in the PDO from *B. leptomitoformis* instead of His214 and Leu212 as in hETHE1 and *Bp*PRF).

As for the second loop, the position of the conserved Phe146 and Gln147 varies in different type I PDOs (Figure 11b and Figure 12), apparently depending on which amino acid residue is at position 148 (Ala, Gln, Arg, Glu, or Asn). It is interesting to note the difference in the position of the side groups of the polar (glutamine) and hydrophobic (phenylalanine) residues in this loop in different PDOs. In hETHE1 and *Mx*PDOI, the polar group of Gln146 is involved in the formation of the active site, while the hydrophobic side group of Phe145 is turned aside. However, in the PDO of *B. leptomitoformis,* Phe146 makes a hydrophobic contribution to the active site surface, while the charged group of Gln147 participates in the formation of the dimer interface. A similar loop arrangement is observed in the structure of the active site of *Bp*PRF. The rotation of the phenyl group of phenylalanine can affect the location of substrate/product in the active site of the enzyme.

The second coordination sphere of the iron ion in the PDO of *B. leptomitoformis* differs from other known type I PDO structures by a single residue (His117), with His replaced by Cys in hETHE1 and *Bp*PRF (Figure 11b) or Ser in *Mx*PDOI (Figure 12). The His117 side group, unlike cysteine and serine in the corresponding position, displaces water molecules located in the internal cavity of the enzyme, which can stabilize the hydrogen bonding network near the active center and affect the enzyme activity.

Thus, the structural analyses demonstrate that there are differences both in the intermonomers’ interface and in the substrate-binding pocket geometry of PDO *B. leptomitoformis* compared with other type I PDOs.

The data obtained suggest that the reaction of sulfur oxidation to sulfite in *B. leptomitoformis* D-402 is catalyzed by persulfide dioxygenase similar to the mechanism described for the heterotrophic bacterium *Cupriavidus pinatubonensis* JMP134 (Figure 13). The thiosulfate formed as an intermediate is used by *B. leptomitoformis* during lithotrophic growth. In this process, the oxidation of thiosulfate occurs with the participation of the Sox-system to generate energy.

The main difference between *B. leptomitoformis* compared to *C. pinatubonensis* JMP134 is in the transformation of thiosulfate, or rather in its oxidation products. In the first case, a branched Sox-system functions, without SoxCD proteins, resulting in the formation of elemental sulfur and sulfate. In the second case, the full Sox-system functions and only sulfates are formed. In a recent study, cells of the mutant strain of *C. pinatubonensis* JMP134, in which the *soxCD* gene was deleted, produced sulfane sulfur, which was almost immediately oxidized with PDO participation [34]. Due to the fact that the sulfur in *B. leptomitoformis* is enclosed in a protein envelope, there is no need for its urgent removal to avoid toxic effects on the cell [2]. Therefore, bacteria can store it as an energy reserve for a long period to re-incorporate it into energy metabolism when needed.

## 4. Materials and Methods

### 4.1. Composition of Nutrient Media

The *B. leptomitoformis* D-402 strain was cultured under different conditions, which were based on the following mineral composition medium, g L^−1^: NaNO_3_ (0.620); NaH_2_PO_4_ (0.125); CaCl_2_·2H_2_O (0.030); Na_2_SO_4_ (0.500); KCl (0.125); MgCl_2_·6H_2_O (0.050). The pH of the medium was adjusted to 7.0 before sowing. Before culturing, 1 mL each of vitamin (thiamine (B_1_), biotin (B_7_) and cobalamin (B_12_)) and 1 mL of trace elements were added to the medium [43]. The culture was incubated at 27 °C.

For chemolithoautotrophic growth in the presence of reduced sulfur compounds, sodium thiosulfate (1.0 g/L Na_2_S_2_O_3_·5H_2_O) or sodium sulfide (0.6 g/L Na_2_S·9H_2_O) was added to the medium as an electron donor for energy metabolism. When cultured with the addition of sodium sulfide, the method of creating gradient media was used. An agar column at a concentration of 15 g/L containing 0.6 g/L Na_2_S·9H_2_O was used as the bottom layer; nutrient medium of the above composition was used as the top layer. Vials with a capacity of 0.5 L were used for cultivation. The height of the bottom layer was 3–4 cm, and the height of the top layer was 7–8 cm.

Chemolithoautotrophic growth in the presence of elemental sulfur was initiated by adding inoculum from the medium containing sodium sulfide as an energy source to the starvation mineral medium with only 0.5 g/L NaHCO_3_ as an inorganic carbon source. The starvation mineral medium contained only a basic mineral composition and NaHCO_3_ as an inorganic carbon source. When such medium was inoculated with *B. leptomitoformis* D-402 cells, elemental sulfur located intracellularly acted as an electron donor for energy metabolism. To remove trace amounts of the energy substrate, two successive passages were performed. Under these conditions, by the second passage, the only possible energy substrate or its precursor is intracellular sulfur. The absence of traces of sodium sulfide in the medium was confirmed using iodometric titration [44]. Intracellular sulfur was observed visually.

To create chemolithoheterotrophic culturing conditions, Na_2_S_2_O_3_·5H_2_O and lactate were added to the mineral medium at concentrations of 1.0 and 0.25 g/L, respectively. Chemoorganoheterotrophic growth of *B. leptomitoformis* D-402 was initiated by adding 0.25 g/L lactate to the mineral medium.

### 4.2. Genome Annotation and Phylogenetic Analysis

Gene search and annotation were carried out using the RAST server 2 [45], followed by manual correction of the annotation by comparing the predicted protein sequences with the National Center for Biotechnology Information (NCBI) databases.

The search for PDO homologs was performed using NCBI BLASTP (http://blast.ncbi.nlm.nih.gov/Blast.cgi, accessed on 10 May 2023). The obtained amino acid sequences were aligned using ClustalW software for multiple sequence alignment. The phylogenetic tree of homologous proteins was constructed by the neighbor-joining method [46] using p-distance in the MEGA 11 program [47]. The fidelity values of branching nodes were evaluated using bootstrap analysis by constructing 1000 alternative trees. Subcellular localization of proteins was predicted using the PSORTB v3.0 tool (http://www.psort.org/psortb/, accessed on 15 May 2023). The presence of signal peptides was predicted using Signal P v.5.0 (https://services.healthtech.dtu.dk/service.php?SignalP-5.0, accessed on 15 May 2023) and InterPro (https://www.ebi.ac.uk/interpro/, accessed on 15 May 2023).

### 4.3. Sample Preparation for Proteomic Analysis

Samples were ultrasonicated on ice in pulse mode for 30 s in triethylammonium bicarbonate (TEAB) buffer containing 4% SDS. The homogenate was centrifuged at 14,000× *g* for 10 min at 10 °C. Protein concentration was determined by the Bradford method on an Evolution 260 Bio spectrophotometer (Thermo Fisher Scientific, Waltham, MA, USA). Sample preparation was carried out by suspension trapping on an S-trap filter using 5% SDS [48].

### 4.4. High-Performance Liquid Chromatography in Combination with Tandem Mass Spectrometry (HPLC-MS/MS)

The mass spectrometric analysis of samples after hydrolysis was performed using an Ultimate 3000 RSLCnano chromatography HPLC system (Thermo Fisher Scientific, Waltham, MA, USA) coupled to a Q-Exactive HF-X mass spectrometer (Thermo Fisher Scientific, Waltham, MA, USA). Chromatographic separation of peptides was performed on an analytical reverse phase column Acclaim Pepmap^®^ C18 of 15 cm in length and with an inner diameter of 75 μm (C₁₈-reversed phase silica gel was used as packing material (particle size 3 μm, pore size 10 nm); Thermo Fisher Scientific, Waltham, MA, USA) in gradient elution mode. The gradient was formed with mobile phase A (0.1% formic acid in deionized water) and mobile phase B (80% acetonitrile, 0.1% formic acid in deionized water) from 5% to 30% for 70 min at a flow rate of 0.4 μL/min followed by equilibration of the chromatographic system at initial gradient conditions (A:B = 2:98) for 8 min.

The mass spectrometric analysis was performed in four replicates on a Q Exactive HF mass spectrometer (Thermo Fisher Scientific, Waltham, MA, USA) in positive ionization mode using a nanoelectrospray ionization source (nESI). Mass spectra were obtained with a resolution of 60,000 (MS) and 15,000 (MS/MS) in the range of *m*/*z* 375–1300 (MS) and 200–2000 (MS/MS), respectively. An isolation threshold of 50,000 counts was determined for precursor selection and the top 10 precursors were selected for fragmentation with high-energy collision-induced dissociation at 29 NCE and an accumulation time of 100 ms. Precursors with a charged state of +1 were discarded and all measured precursors were excluded from the measurement for 10 s. The maximum accumulation time for precursor ions was 50 ms and 150 ms for fragment ions.

The proteins were identified using the MaxQuant v.1.6.17.0 program with the Andromeda search algorithm [49]. The UniProt (Swiss-prot) database of *Beggiatoa leptomitoformis* D-402 was used for protein identification. The following parameters were determined: the proteolytic cleavage enzyme was trypsin; the mass accuracy of monoisotopic peptides was ±5 ppm; the mass accuracy of the MS/MS spectra was ±0.01 Da; and there was the possibility of missing two cleavage sites by trypsin. Modification of cysteine by chloroacetoamide was considered as a fixed peptide modification, and methionine oxidation and N-terminal acetylation were considered as variable modifications. An FDR (False Discovery Rate) value of no more than 1.0% was used for validation of PSMs (Peptide-Spectrum Matches), peptide identification, and protein identification. Proteins for which at least two peptides were detected were considered as reliably identified. The protein content label-free quantification was based on iBAQ and LFQ.

Statistical analysis of the data obtained during identification was performed in the Perseus v.1.6.15.0 program. The data were preliminarily filtered to select the most significant points: possible contaminant proteins and unreliable identified proteins were removed.

To isolate significant differences between sample proteins, the list of proteins was further filtered; proteins that occurred in at least two repetitions of the analysis out of four were retained.

### 4.5. Cloning of the PdoI Gene

Genomic DNA was isolated using the diaGene Kit (Dia-M, Moscow, Russia) according to the manufacturer’s instructions. The gene of persulfide dioxygenase was identified in the annotated genome sequence of *B. leptomitoformis* and analyzed using Blast (https://blast.ncbi.nlm.nih.gov/Blast.cgi, accessed on 10 May 2023) and InterPro (https://www.ebi.ac.uk/interpro/, accessed on 15 May 2023). The full-length gene of PDO was amplified from genomic DNA by PCR using BlitzMasterMix (Belbiolabs, Moscow, Russia). The sequences of the forward PdoIFe_Nde and the reverse PdoIRe_Xho_22 primer were AGTCATATGATTTTTCAACAATTATTTGAGTCTAG and ATATCTCGAGCACCAGTGTTTCACATAACTG, respectively. The PCR amplification program was as follows: 98 °C, 30″ + [(98 °C, 10″ + 50 °C, 30″ + 72 °C, 1′) × 35] + 72 °C 2′. The PCR product of the appropriate size was purified using the diaGene Kit (Dia-M, Moscow, Russia), and the gene sequence was confirmed by sequencing.

To clone the gene into the pET-22b(+) expression vector at the NdeI and XhoI restriction endonuclease sites, the plasmid and PCR product were treated with restriction endonucleases (Thermo Fisher Scientific, Waltham, MA, USA) according to the manufacturer’s instructions. Then, the ligation reaction of the vector and amplicon was carried out at +15 °C using T4 DNA ligase (Thermo Fisher Scientific, Waltham, MA, USA). Competent cells of *E. coli* DH5a were transformed with a ligation mixture containing the *pET22b::pdoI* construct. Insertion plasmids were obtained from the overnight culture of transformants. The sequence of the inserted *pdoI* was verified for the absence of mutations by sequencing.

### 4.6. Recombinant Expression and Purification

For the production of recombinant PDO, the strain *E. coli* BL21-CodonPlus (DE3) was transformed with the *pET22b::pdoI* plasmid and grown at 37 °C with agitation at 250 rpm to a cell density of 0.6–0.8 (A600). Then, 0.9 mM isopropyl-β-D-thiogalactopyranoside was added to the culture medium and the cells were incubated for 18 h at 16 °C with agitation at 100 rpm. Cells were collected by centrifugation at 4000× *g* for 20 min, suspended in 35 mL of 50 mM KPi buffer, pH 7.5, containing 0.5 M NaCl, 1 mM imidazole (buffer 1) with the addition of protease inhibitors (1 mM alpha-aminocaproic acid, 1 mM PMSF, and 1 mM benzamidine) and disrupted by sonication. Cell debris was removed by centrifugation (90 min at 12,000× *g*) and protein was purified by affinity chromatography on a HisTrap 5 mL column (GE Healthcare, Chicago, IL, USA). Cell extract was loaded onto a HisTrap column equilibrated with Buffer 1, then washed with five volumes of Buffer 1 and then washed with five volumes of Buffer 2 (50 mM KPi buffer, 0.5 M NaCl, 50 mM imidazole, pH 7.5). Fractions containing the enzyme were eluted with Buffer 3 (50 mM KPi buffer, 0.5 M NaCl, 300 mM imidazole, pH 7.5). After the chromatography stage, the protein was dialyzed against 50 mM KPi buffer (pH 7.5). The concentration of the protein was determined using the molar extinction at 280 nm (ε = 15,040 M^−1^ × cm^−1^) calculated from the protein sequence using the Vector NTI Program (Life Technologies, Carlsbad, CA, USA). The molecular mass of enzyme was determined by 12% sodium dodecyl sulfate-polyacrylamide gel electrophoresis (SDS-PAGE) according to Laemmli, 1970 [50]. 

### 4.7. PDO Activity Assay

PDO activity was analyzed using the method of Suzuki, 1965 [51], based on the detection of oxygen consumption [17] with modifications. The O_2_ consumption rate was measured with a Clarke-type electrode in a 1 mL cell (Hansatech Instruments Ltd., Norfolk, UK) at 35 °C (unless otherwise indicated) in the reaction media containing a mixture of 50 mM KPi buffer (pH 7.0) and GSSH in a 1:1 ratio. GSSH was produced by mixing equal volumes of 17 mM reduced glutathione in distilled water and a saturated sulfur solution, containing about 17 mM elemental sulfur [52]. Before the measurement, the reaction mixture was pre-incubated for 1 min in the cell covered with a cap; after that, the non-enzymatic change in the O_2_ level was recorded for 140 s. The enzymatic reaction was initiated by the PDO addition through the channel in the chamber cap with a gel loading tip. Before measurements, the GSSH mixture and PDO were kept on ice. For calculation of the maximum O_2_ consumption rate, a linear part of the curve was selected. The concentration of the enzyme in the cell was 30 µg. The calibration of the electrode was performed by adding a few grains of dithionite, and the concentration of dissolved O_2_ was assumed as 219 µM mL^−1^. The O_2_ consumption rate was determined as µmol O_2_ mg^−1^ h^−1^. One unit (U) of PDO activity was defined to consume 1 nmol O_2_ mg^−1^ min^−1^ [16].

### 4.8. Biochemical Characteristics of PDO

The effect of pH and temperature on PDO activity and stability was tested using GSSH as a substrate. The pH optimum of PDO activity was measured in the range of 6.0 to 8.0 in increments of 0.5 units at 35 °C in 50 mM KPi buffer. The optimum temperature of enzyme activity was measured from 25 °C to 55 °C in increments of 5 °C in 50 mM KPi buffer (pH 7.0). The thermal stability of PDO was determined by measuring the residual enzyme activity after one hour of incubation at 30 °C, 40 °C, 45 °C, and 50 °C in the same buffer at pH 7.0. The stability of the enzyme at different pH values was evaluated by incubating the protein in 50 mM KPi buffer in cold conditions for a week over the pH range of 6.0 to 11.0 by increments of 1.0 unit.

Residual activity assay was determined in 50 mM KPi buffer (pH 7.0) at 35 °C using GSSH as a substrate.

The inhibition of enzyme activity was evaluated using metal ions (Cu^2+^, Zn^2+^, Mn^2+^, Mg^2+^, Fe^3+^) and EDTA by assaying PDO activity as described above in the presence of each reagent at a concentration of 1 mM in 50 mM KPi buffer (pH 7.0) at 35 °C [11].

### 4.9. Crystallization

Crystallization experiments were performed at 22 °C using the hanging-drop vapor-diffusion method on siliconized glass cover slides in Linbro plates (Molecular Dimensions Ltd., Sheffield, UK). Crystallization drops were made by mixing 1 μL of protein solution with a concentration of 10 mg/mL in 50 mM KPi buffer (pH 7.0) containing 100 mM NaCl and 1 μL of reservoir solution. Crystals of PDO were obtained in 8% *w*/*v* PEG 4000, 0.2 M imidazole malate, pH 6.0 (condition # 13 of Stura Footprint Screen-2, Molecular Dimensions Ltd., Sheffield, UK). Prior to flash freezing, a single crystal was soaked in a cryo solution consisting of 25.5% *w*/*v* PEG 4000, 15% glycerol, 0.085 M sodium citrate, pH 5.6, 0.17 M ammonium acetate (Condition # 9 of Crystal Screen Cryo, Hampton Research, Aliso Viejo, CA, USA).

### 4.10. Crystallography

Diffraction data from the PDO crystal were collected on a beamline BL18U1 at SSRF, China. The reflection dataset was processed using XDS [53]. The structures were determined using molecular replacement with Phaser [54], with the hETHE1 structure, determined at a 2.6 Å resolution (pdb id 4CHL), being used as a search model. The water molecules and metal ions were removed from the model. The initial model was subjected to crystallographic refinement with REFMAC5 [55]. The manual rebuilding of the model was carried out in Coot [56]. The final cycle, with occupancy refinement, was performed in Phenix [57]. The data and refinement statistics are summarized in Table 2. The atom coordinates and structure factors have been deposited in the Protein Data Bank (http://wwpdb.org/, accessed on 26 April 2024). Figures were prepared using PyMOL (http://www.pymol.org/, accessed on 20 October 2023) [58].

### 4.11. Gene Expression Analysis

Gene expression was analyzed during the chemolithoautotrophic growth of *B. leptomitoformis* D-402, when either sodium sulfide or intracellular sulfur was used as an electron donor for energy metabolism, and during organoheterotrophic growth as a control. The gene of DNA gyrase subunit B (*gyrB*) and the 16S rRNA gene (*rrs*) were used as reference genes for analysis.

RNA was isolated using the ExtractRNA reagent (Evrogen, Moscow, Russia) in accordance with the manufacturer’s protocol. RNA quality was evaluated by electrophoresis on a 1.1% agarose gel with 2.2 M formaldehyde added. RNA concentration was measured using an HS Equalbit RNA assay kit (Vazyme, Nanjing, China) on a Fluo-200 fluorometer (Allsheng, Hangzhou, China). Then, 2000 ng of RNA was reverse transcribed using M-MulV (SybEnzyme, Moscow, Russia) according to the manufacturer’s protocol. Quantitative RT-PCR was performed using SYBR Green I on a Bio-Rad CFX96 touch real-time PCR detection system (Bio-Rad, Hercules, CA, USA).

A temperature gradient was used to find optimal amplification conditions. The resulting program for *soxAXBYZ* genes of the Sox-system included 95 °C, 5′ + [(95 °C, 10″ + 58 °C, 20″ + 72 °C, 15″) × 35]; for the *rrs*, *gyrB*, and *pdo* genes, the program included 95 °C, 5′ + [(95 °C, 20″ + 60 °C, 20″ + 72 °C, 15″) × 35]. Fragments of genes were amplified using the primers 5′–TGGGCTGTACCAGAAGTAGGT–3′ and 5′–GTTACGACTTCACCCCAGTCA–3′ for the *rrs* gene; 5′–ACGCATTTATTCAACGGGGC–3′ and 5′–ACGGCGGGCTTCATTCATTA–3′ for the *gyrB* gene; 5′–TTTCCCACCTATCGCACCAG–3′ and 5′–TCGAATAGCGACACCACACC–3′ for *soxAX*; 5′–AACACGATGCACAAATCGCC–3′ and 5′–TGCAGGTTTGGTCTAGGACG–3′ for *soxB*; 5′–TAAAATGGGTGGGACGGGTG–3′ and 5′–CCACAACCGCCAATAGTCAC–3′ for *soxY*; 5′–ATACGGGCGAGTTGATTCCTG–3′ and 5′–GCTCCAAATGGCACTCATCAC–3′ for *soxZ*; and 5′–TTGCCCAAGAAAAACAGCGT–3′ and 5′–TTTGCGCGGATTTGGTGTTT–3′ for *pdo*. All primers were designed with PrimerBLAST (http://www.ncbi.nlm.nih.gov/tools/primer-blast, accessed on 8 June 2023).

### 4.12. Quantitative Method for the Determination of Elemental Sulfur, Thiosulfate, and Sulfate

Sulfate and thiosulfate were assayed using a Stayer HPLC chromatograph (Aquilon, Moscow, Russia) equipped with a conductivity detector. Separation was operated isocratically at 35 °C on a Dionex IonPac AS22 column (Thermo Fisher Scientific, Waltham, MA, USA) with 4.5 mM sodium carbonate/1.4 mM sodium bicarbonate buffer plus 10% of methanol with a flow rate of 1.0 mL/min. The column (250 × 4 mm, particle size 6 µm) had a polyvinylbenzyl ammonium matrix cross-linked with 55% of divinylbenzene and alkanol quaternary ammonium as a functional group. Before injection, the samples of microbial suspensions were freed from the cells by centrifugation at 15,000× *g* and 15 min.

Changes in the content of sulfur inclusions in *B. leptomitoformis* D-402 cells were observed visually using an Olympus CX31 HD Digital microscope (Olympus, Tokyo, Japan) equipped with objectives of the AmScope Infinity Plan 100X/1.25 Oil with phase contrast. The culture of strain D-402 was analyzed by microscope in at least 30 fields of view. Sulfur globules were defined as highly refractive intracellular inclusions of a spherical shape.

## 5. Conclusions

The freshwater members of the genus *Beggiatoa*, *B. leptomitoformis* and *B. alba*, usually live in places where there is no constant influx of hydrogen sulfide, and, as a consequence, they have adapted to successfully tolerate the absence of other reduced sulfur compounds by performing cyclic transformations of sulfane sulfur, in which GSH, SO_3_^2−^ and S_2_O_3_^2−^ are intermediates.

Representatives of the genus *Beggiatoa* are mobile due to gliding, and due to this they easily migrate. They can move long distances by the principle of chemotaxis in search of an energy substrate. At the same time, they must somehow store energy inside the cells to survive the period of searching for an energy substrate. It is very likely that such a storage substance is elemental sulfur. Representatives of the genus *Beggiatoa* have adapted to spend sulfur sparingly. First, they slowly oxidize it chemically, accumulating the necessary substrate intracellularly (thiosulfate), without energetic output, and then, under unfavorable conditions, they activate enzymatic systems to obtain energy. One such process is the dissimilatory oxidation of thiosulfate involving the branched Sox-system, which produces sulfate and the re-accumulation of intracellular elemental sulfur. This cyclic transformation of sulfur has the character of a supporting metabolism for representatives of the genus *Beggiatoa*.

## Figures and Tables

**Figure 1 ijms-25-10962-f001:**
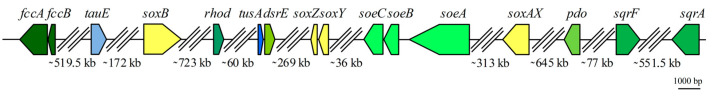
Organization of genes encoding enzymes for dissimilatory sulfide oxidation in the genome of *B. leptomitoformis*. *fccA*—cytochrome c subunit of flavocytochrome c sulfide dehydrogenase (EC 1.8.2.3; AUI67863.1); *fccB*—sulfide dehydrogenase (flavocytochrome c) flavoprotein chain (EC 1.8.2.3; AUI67862.2); *tauE*—sulfite exporter TauE/SafE family protein (AUI67490.1); *soxB*—thiosulfohydrolase SoxB (EC 3.1.6.20; AUI67367.1); *rhd*—rhodanese (sulfur transferase) (EC 2.8.1.1; AUI70268.1); *tusA*—sulfur carrier protein TusA (AUI70013.1); *dsrE*—DsrE/DsrF/DrsH-like family protein (AUI70012.1); *soxZ*—thiosulfate oxidation carrier complex protein SoxZ (EC 1.8.2.6; AUI69817.1); *soxY*—thiosulfate oxidation carrier protein SoxY (EC 1.8.2.6; AUI69816.1); *soeABC*—membrane-bound cytoplasmic sulfite:quinone oxidoreductase SoeABC (EC 1.8.5.6; AUI70655.1, AUI69788.1, AUI69789.1); *soxAX*—sulfur oxidation c-type cytochrome (EC 1.8.2.3; AUI69545.1); *pdo*—persulfide dioxygenase (EC 1.13.11.18; AUI69048.1); *sqrF*—sulfide:quinone oxidoreductase type VI SqrF (EC 1.8.5.4; AUI68992.1); *sqrA*—sulfide:quinone oxidoreductase type I SqrA (EC 1.8.5.4; AUI68548.1).

**Figure 2 ijms-25-10962-f002:**
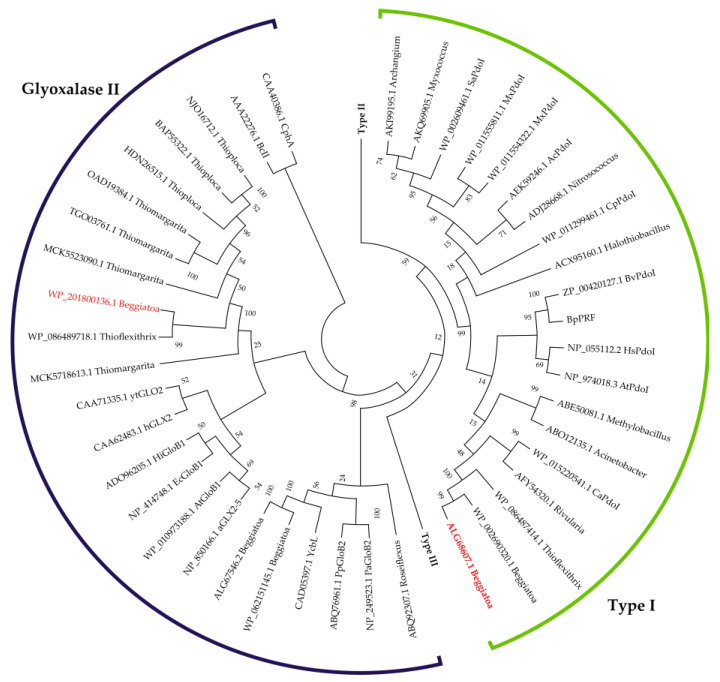
Phylogenetic tree based on the predicted amino acid sequences of PDOs. Clustering sequences were deduced using the neighbor-joining method. The amino acid sequences of related proteins from the superfamily of metallo-β-lactamase, glyoxalase II proteins were used as an outgroup. Phylogenetic analysis was performed based on a representative selection of amino acids sequences of PDOs from the study of Xia et al. [14]. The phylogenetic analysis included 110 amino acid sequences, including sequences for members of the family *Beggiatoaceae* identified by BLASTP searches in this study. Two amino acid sequences, presumably PDOs, found in this study in *B. leptomitoformis* are highlighted in red font. The type I PDO cluster is shown in a green semicircle, and the glyoxalase II proteins cluster in dark purple; type II and III PDOs are condensed within two separate branches. The protein accession numbers are indicated on the tree.

**Figure 3 ijms-25-10962-f003:**
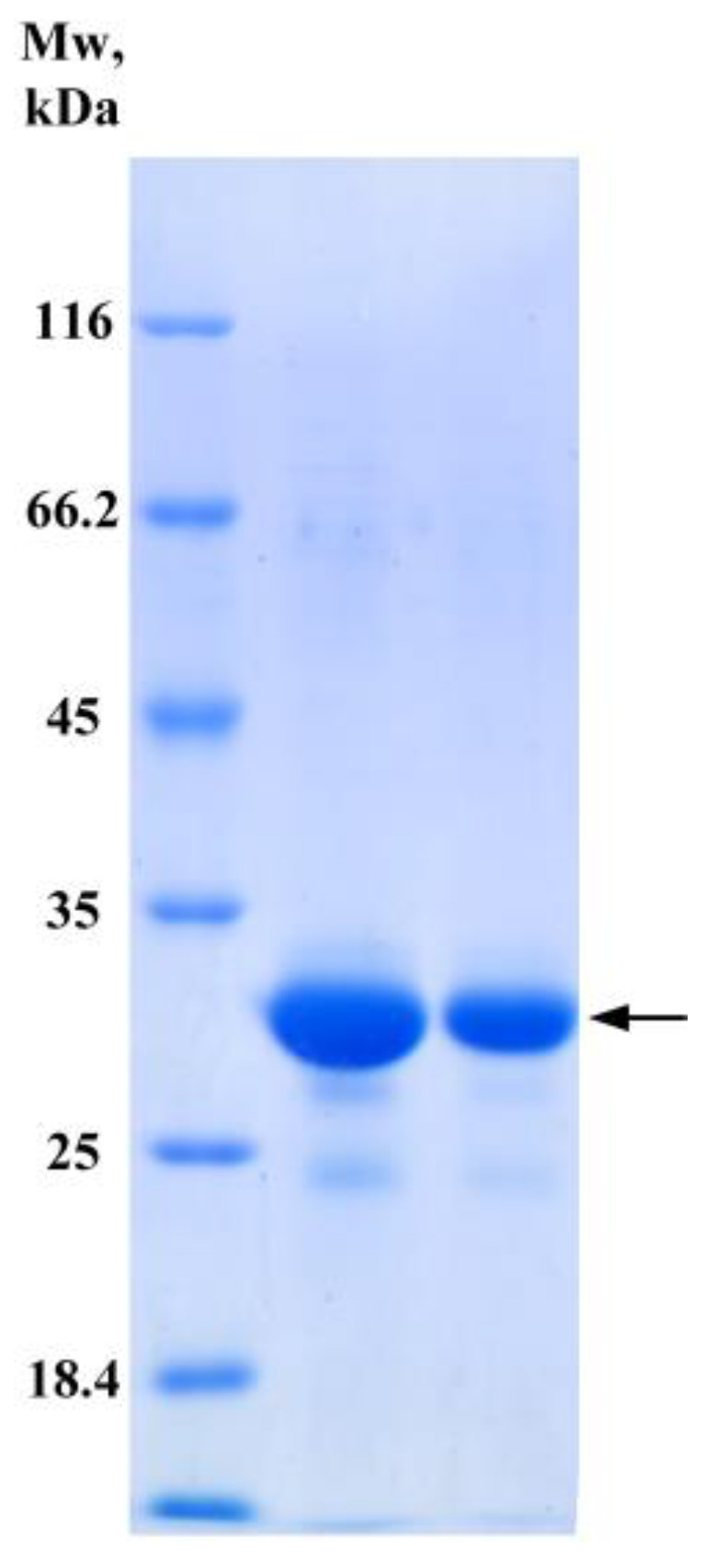
SDS-PAGE of recombinant PDO from *B. leptomitoformis*. The concentrating gel is 5% acrylamide, the separating gel is 12% acrylamide; colored with Coomassie brilliant blue R250. Mw, molecular weight marker (kDa) (Thermo Fisher Scientific, Waltham, MA, USA). The arrow shows the band with the target protein PDO.

**Figure 4 ijms-25-10962-f004:**
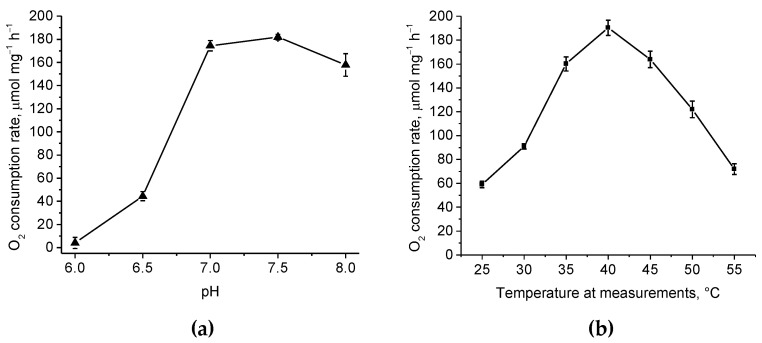
The effects of pH (**a**) and temperature (**b**) on the activity of recombinant PDO of *B. leptomitoformis* in the presence of GSSH. The bars on the curves show the standard deviations of the triplicate measurements.

**Figure 5 ijms-25-10962-f005:**
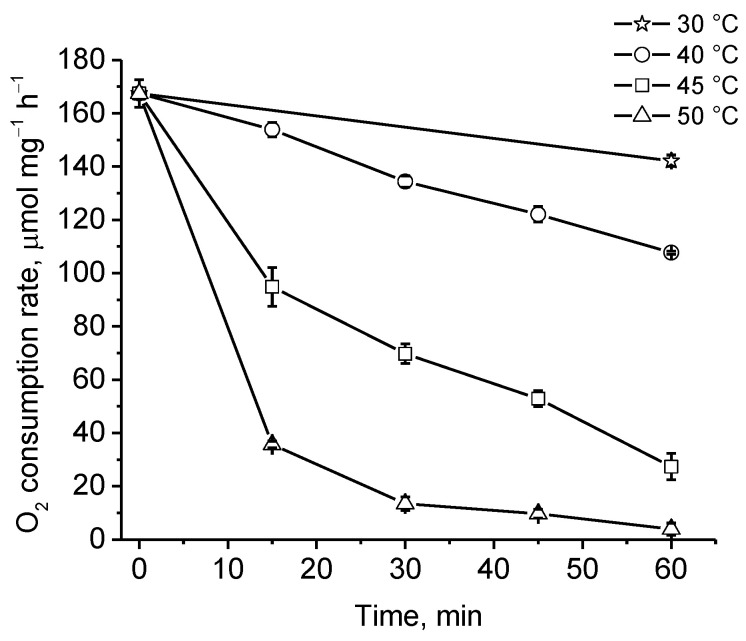
Stability assay of recombinant PDO of *B. leptomitoformis* at different temperatures in the presence of GSSH. Residual activity assay was determined in 50 mM KPi buffer (pH 7.0) at 35 °C. The bars on the curves show the standard deviations of the triplicate measurements.

**Figure 6 ijms-25-10962-f006:**
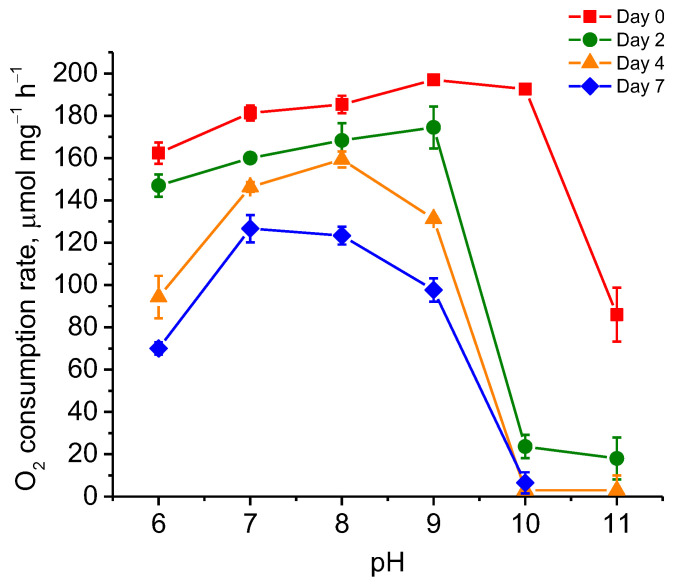
Stability assay of recombinant PDO of *B. leptomitoformis* at different pH values in the presence of GSSH. Residual activity assay was determined in 50 mM KPi buffer (pH 7.0) at 35 °C. The bars on the curves show the standard deviations of the triplicate measurements.

**Figure 7 ijms-25-10962-f007:**
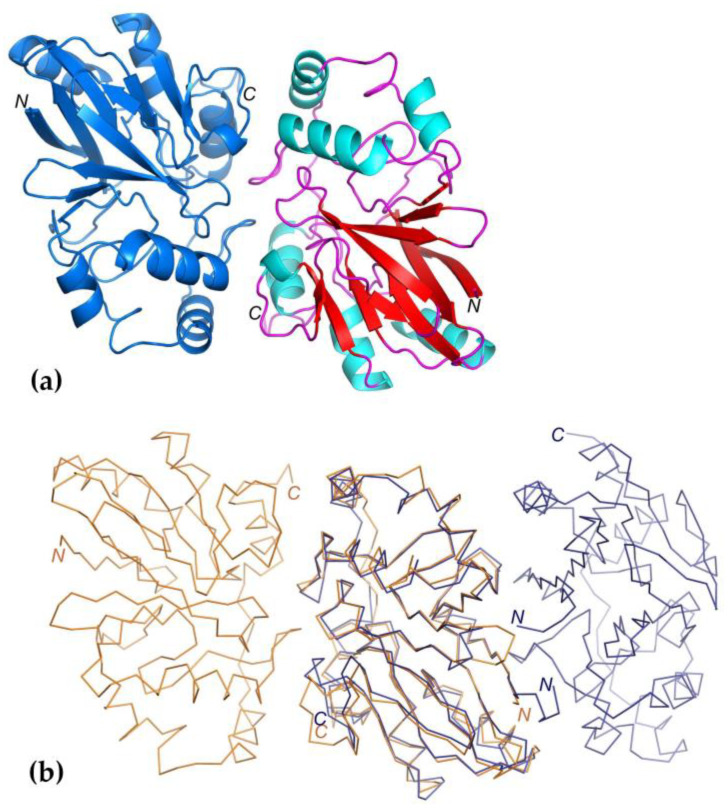
General view of the crystal structure of PDO from *B. leptomitoformis* (pdb id 8ZBD) and superimposed structures of PDO from *B. leptomitoformis* and hETHE1 (pdb id 4CHL). (**a**) *B. leptomitoformis* PDO dimer; secondary structure elements in one of the monomers are shown in different colors (β-sheets in red, α-helixes in blue, and loops in purple). (**b**) From left to right: Cα-atom trajectories of PDO monomers from *B. leptomitoformis* in dimer (orange) and Cα-atom trajectories of hETHE1 monomers (blue).

**Figure 8 ijms-25-10962-f008:**
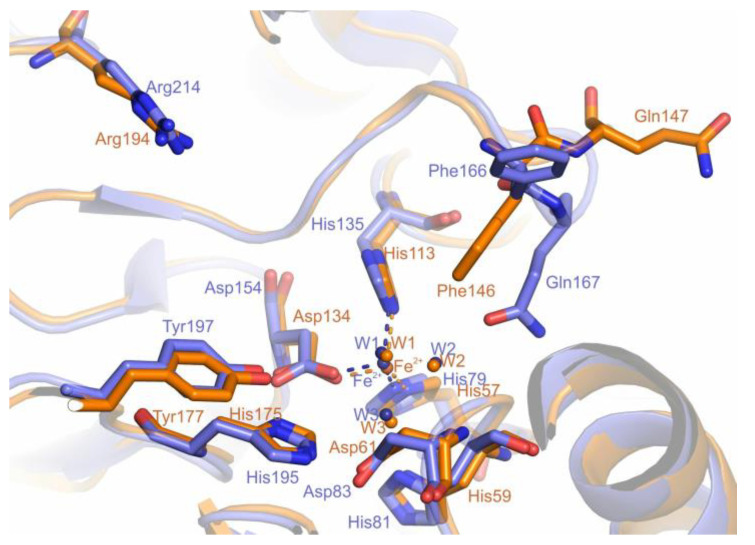
Superposition of *B. leptomitoformis* PDO active site structures (orange) and hETHE1 (blue). The iron ions and water molecules (W1, W2, W3) which coordinate (dashed lines) to the iron are shown as spheres.

**Figure 9 ijms-25-10962-f009:**
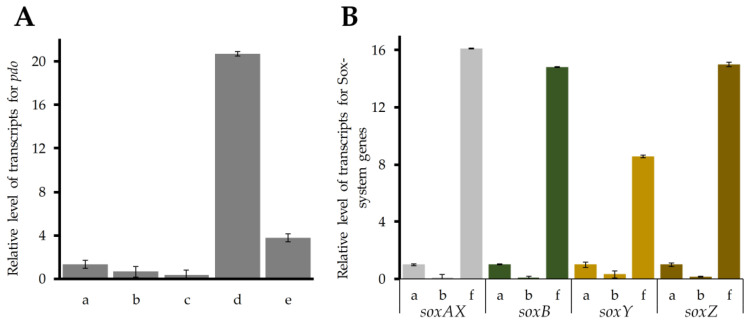
Expression levels of the *pdo* gene (**A**) and Sox-system genes (**B**) in *B. leptomitoformis* D-402 during chemolithoautotrophic growth in the presence of Na_2_S, during the exponential growth phase (3 days) (a), organoheterotrophic growth, during the exponential growth phase (3 days) (b), chemolithoautotrophic growth in the presence of intracellular sulfur, during 4 days of growth (c), during 7 days of growth (d), and during 24 days of growth (e), and chemolithoautotrophic growth in the presence of intracellular sulfur, on day 18 of growth (f).

**Figure 10 ijms-25-10962-f010:**
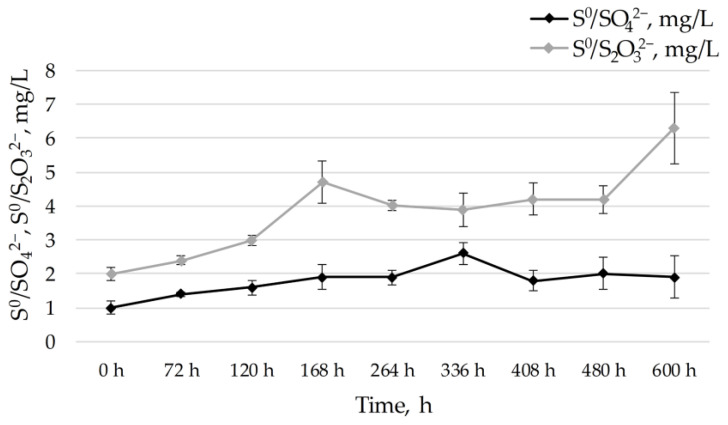
Intermediates detected during the chemolithoautotrophic growth of *B. leptomitoformis* D-402 in the presence of intracellular sulfur. Thiosulfate concentration is shown by gray curves and sulfate concentration by black curves. The bars on the curves show the standard deviations of triplicate chemical measurements. The diagram shows nine points representing 600 h (25 days) of incubation of the strain D-402 in the presence of intracellular elemental sulfur and CO_2_ alone.

**Figure 11 ijms-25-10962-f011:**
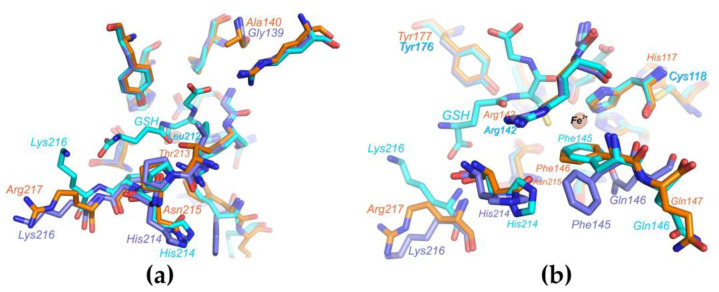
Superposition of the active site structures of PDO from *B. leptomitoformis* (orange), hETHE1 (cyan), and *Bp*PRF (lilac). Panels (**a**,**b**) show different views of the active sites of the three type I PDOs.

**Figure 12 ijms-25-10962-f012:**
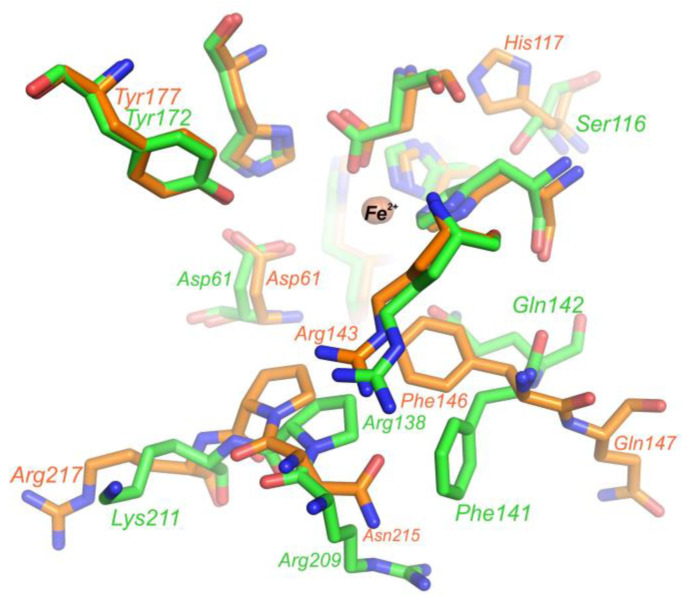
Superposition of the active site structures of PDO *B. leptomitoformis* (orange) and *Mx*PDOI (green). The iron ion is shown as a sphere.

**Figure 13 ijms-25-10962-f013:**
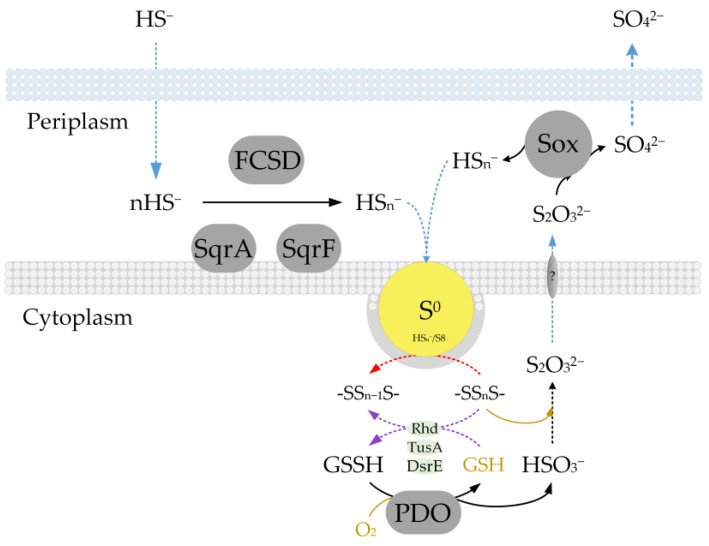
A hypothetical of sulfide oxidation scheme for *B. leptomitoformis* D-402. The black dashed line shows non-enzymatic reactions, the red dashed line shows reactions whose mechanism is still unknown, and the purple dashed line is not shown for *B. leptomitoformis*. FCSD, flavocytochrome c sulfide dehydrogenase; PDO, persulfide dioxygenase; SQR, sulfide:quinone oxidoreductase; GSH, glutathione; GSSH, glutathione persulfide. The figure was adapted from Xin et al., 2020 [15], and Dahl, 2020 [26], under the terms of the Creative Commons Attribution 4.0 International License.

**Table 1 ijms-25-10962-t001:** Effect of metal ions and chemical reagents on the activity of recombinant PDO of *B. leptomitoformis*.

	Activity, %	SD
Control	100.0	2.9
Cu^2+^	23.7	1.5
Zn^2+^	11.2	3.2
Mn^2+^	83.3	4.8
Mg^2+^	99.6	3.7
Fe^3+^	47.2	1.4
Co^2+^	47.4	6.7
Ni^2+^	20.5	6.4
EDTA	92.5	2.3

**Table 2 ijms-25-10962-t002:** Crystallographic data collection and refinement statistics.

Data Collection	
Wavelength (Å)	0.97853
Resolution range (Å)	50.00–2.33 (2.47–2.33)
Space group	P4_1_2_1_2
Cell parameters	
a, b, c (Å)	66.9, 66.9, 245.0
α, β, γ (◦)	90.0, 90.0, 90.0
Collection temperature (K)	100
Total reflections	248,845 (38,956)
Unique reflections	24,873 (3845)
R_merge_ (%)	25.7 (173.2)
Multiplicity	10.00 (10.13)
Completeness (%)	99.6 (97.9)
Mean I/sigma (I)	10.13 (1.70)
Wilson B-factor (Å^2^)	31.7
CC_1/2_	0.99 (0.72)
Refinement	
Resolution range	46.51–2.33 (2.42–2.33)
Reflections used in refinement	24,834 (2491)
Reflections used for R-free	1205 (129)
R-work, %	18.92 (26.62)
R-free, %	24.60 (33.19)
RMSD bond lengths (Å)	0.008
RMSD bond angles (◦)	0.984
Ramachandran favored (%)	98.28
Ramachandran allowed (%)	1.72
Ramachandran outliers (%)	0.00
Average B-factor (Å^2^)	33.88
Macromolecules	33.99
Ligands	33.28
Solvent	29.91
PDB ID	8ZBD

## Data Availability

Data sharing is not applicable.

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
