# Peer review of "Mechanism of Intracellular Elemental Sulfur Oxidation in Beggiatoa leptomitoformis, Where Persulfide Dioxygenase Plays a Key Role"

_ijms, 2024, doi:10.3390/ijms252010962_

Round 1
Reviewer 1 Report
Comments and Suggestions for Authors
The manuscript describes the aerobic bacterium Beggiatoa leptomitoformis which accumulates intracellular sulfur. When the cell requires energy, the sulfur disappears catalyzed by an unknown enzyme. Genomic search using known protein sequences of persulfide dioxygenases led two genes with 30% sequence identity. The phylogenetic analysis showed clear clustering of the derived sequence ALG68607.1 with persulfide dioxygenase type I sequences. The other sequence WP_201800136.1 clustered with glyoxalase. The ALG68607.1 sequence was expressed in E. coli. and the resulting protein was purified and crystalized for the determination of the crystal structure. The enzyme was characterized as persulfide dioxygenase (PDO) with a specific activity of 174 µmol x mg-1 x h-1. Taking the international defined specific activity as µmol x mg-1 x min-1, the specific activity of PDO reduces to 3 µmol x mg-1 x min-1, a low activity for an efficient enzyme! This agrees with the physiological function of the enzyme, just to provide energy for the movement of the cell and not for growth.
Give the complete reaction catalyzed by PDO:
GSSH + O2 + H2O = GSH + SO32- + 2 H+
See: J. Biol. Chem. (2018) 293(32) 12429 –12439
Questions:
Has the glyoxalase WP_201800136.1 also PDO activity?
What is starvation mineral medium?
Line 192: “Nine conservative amino acids were present in the sequence: two histidines, two aspartic acids, one leucine, one cysteine, one glycine, one arginine and one tyrosine coordinating metal atoms.” Mention the proteins in which the amino acids are conserved! Give the amino acid numbers in the sequences.
Line 783: “One such process is the dissimilatory oxidation of thiosulfate involving the branched Sox system, in which there is a re-accumulation of intracellular elemental sulfur and sulfate.” Does this sentence fit to the final conclusion?
Line 461: has B. leptomitoformis compartments? Most bacteria don’t have.
Typos:
Line 49, Schmidt et al. Give the reference number.
Line 117: sulfur
Line 120: use sequence identity instead of homology.
Line 128: For The amino acid
Lines 199 and 201: use molecular mass instead of weight.
Line 325: that can be realized a mechanism
Line 546: removal
Comments on the Quality of English Language
The few incorrect English sentences are listed under "typos".
Author Response
Translator Dear Reviewer,
Dear Reviewer,
Please see the attachment.

Reviewer 2 Report
Comments and Suggestions for Authors
The authors of this research did a thorough work at identifying a possible enzyme at a bioinformatic level and then do a comprehensive characterization of the enzyme at a functional and structural level and at in vitro and at in vivo. Although this research has a lot of data, it is well written and easy to read and interpret.
I pose the authors some minor questions/comments.
1) In nowhere it states that PDO is an iron enzyme, only when the structure is presented and described is when we discover it is an iron-protein. It would also be interesting to show UV-Vis spectra of both reduced and oxidized states.
2) On line 252, you used EDTA to show that the activity still remains. Did you use a stronger chelant (such as dipicolinic acid) to show that iron is crucial for the enzymes’ activity?
3) On line 483, you said that several metal have inhibitory effects, and correctly site references, however, you should write why they have an inhibitory effect. Do they substitute the iron ion at the active site?
4) On line 484 you stated that manganese reduces marginally the proteins’ activity, I disagree that almost 20% is a marginal effect.
5) Figure 8 shows the iron atoms as spheres, but I believe that there is only a single iron atom, the other spheres are probably waters. Can you color them differently or identify them as waters so the readers do not misidentify?
6) In the crystallization section (4.9) is the buffer if the protein 50 mM KPi, pH 7.0? It is not written what is the initial condition prior to crystallization.
This article fits the scope of the Molecular Microbiology Special Issue Current Research on Omics of Microorganisms and I would accept this article with minor revisions.
Author Response
Translator
Dear Reviewer,
Please see the attachment.

Reviewer 3 Report
Comments and Suggestions for Authors
The publication is important because it provides insight into sulfur metabolism. It is worth emphasizing that there is no universal mechanism for sulfur oxidation in bacteria. Additionally, research is often hampered by the bidirectional action of enzymes. This publication seems to be within the scope of journal. However it needs several corrections to be more acceptable for publication.
Fig. 13 Please note that when sulfane sulfur is released from GSSH as HSO3-, GSH is simultaneously formed, which is converted to GSSH by RHD. The last reaction is not visible in this figure. If the authors believe that this reaction is non-enzymatic (black dashed line), why is there Rhd above the arrow?
Legend of fig 9 (line 312) It is not clear after how many days PDO expression was measured in chemolithoautotrophic growth in the presence of Na2S (a) and organoheterotrophic growth (b).
Line 555, Please add information about exactly what vitamins were used.
Line 569 Please add information or references how the iodometric titration was performed.
Line 572 It should be „0.25 g/L” instead of „0.25 g\L”.
Line 603 Please add information about column filling.
Line 711, 712, 763 it should be „mL” and „µL” instead of „ml” and „µl”, respectively. Please check carefuly the whole manuscript and correct evident mistake.
Line 762: I assume the authors meant sodium carbonate / sodium bicarbonate. Please correct evident mistake. Please add information about column parameters.
Author Response

(The authors gave the same response as above.)
